# Emergency Approach to Acute Seizures in Dogs and Cats

**DOI:** 10.3390/vetsci11060277

**Published:** 2024-06-17

**Authors:** George G. Munguia, Aimee C. Brooks, Stephanie A. Thomovsky, Elizabeth J. Thomovsky, Andrea Rincon, Paula A. Johnson

**Affiliations:** Department of Veterinary Clinical Sciences, College of Veterinary Medicine, Purdue University, 625 Harrison St., West Lafayette, IN 47907, USA

**Keywords:** seizure, benzodiazepines, cluster, status epilepticus, levetiracetam, phenobarbital, cats, dogs, veterinary neurology

## Abstract

**Simple Summary:**

Seizures are commonly noted in dogs and cats and should prompt immediate veterinary assessment and care in first time seizure patients or patients with multiple or ongoing seizures. All animals presenting for seizures should be evaluated to ensure they are not in imminent danger of death, to confirm that the episode noted at home was a seizure, and to look for the underlying causes of seizures. In certain situations, animals with seizures require immediate anticonvulsant medications and possibly in-hospital therapy and monitoring. This review presents a stepwise approach for veterinarians evaluating patients presenting for seizures.

**Abstract:**

Seizures are a common presentation seen in small animal practices. Seizures require prompt management including initial interventions for triage, stabilization, and treatment with first-line anticonvulsant (AC) drugs like benzodiazepines. Concurrently, ruling out metabolic or extracranial causes with point-of-care diagnostics can help guide further diagnostics and treatments. Analysis of the history and a physical exam are also necessary to rule out common “look-alikes” that require specific diagnostic workup and treatments. Typically, causes of seizures can be grouped into intracranial and extracranial causes, with the latter being easier to diagnose with commonly available tests. This review presents a systematic approach to the diagnosis and treatment of single seizures, cluster seizures, and status epilepticus in dogs and cats.

## 1. Introduction

Seizures and seizure-like activity are common presenting complaints to both small animal emergency hospitals and general practices. A small animal clinician should be prepared to assess, quickly treat, and note different causes of seizures while also ruling out other “look alike” episodes. The following is a practical guide to approaching canine and feline patients presenting for single-event seizures, cluster seizures, or status epilepticus.

## 2. Patient Presentation for Suspected or Confirmed Seizures

### Initial Assessment and Treatment

As with any emergency patient, it is important to immediately perform triage and assess stability and vital parameters. This includes ensuring the animal is not actively having focal or generalized seizures or suffering from secondary effects of seizures (see below: *Other Considerations for Cluster Seizures and Status Epilepticus*). If the animal is having a seizure at the time of presentation, the priority is to stop the seizure. 

The first-line anti-convulsant (AC) medications of choice to treat seizures are benzodiazepines, including diazepam and midazolam. Ideally, these drugs are given intravenously. However, if venous access is challenging or delayed, intranasal (IN) and/or per rectum (PR) routes of administration of benzodiazepines are recommended. As described in the upcoming Section *Causes of Seizures*, metabolic conditions such as hypoglycemia may produce seizures. Thus, assessment and treatment of metabolic diseases (Table 1) is also integral in the control of seizure activity. Table 2 shows common AC drug options and doses, and Figure 1 describes an algorithm for the administration of AC drugs to the actively seizuring patient. 

Patients presenting for seizures should be actively monitored. This includes keeping the patient in a visible kennel so that even subtle changes can be observed. Cameras can be used in lieu of a direct line of sight. Some hospitals also tie small bells to the pet’s collar to alert staff to movements consistent with seizure.

**Figure 1 vetsci-11-00277-f001:**
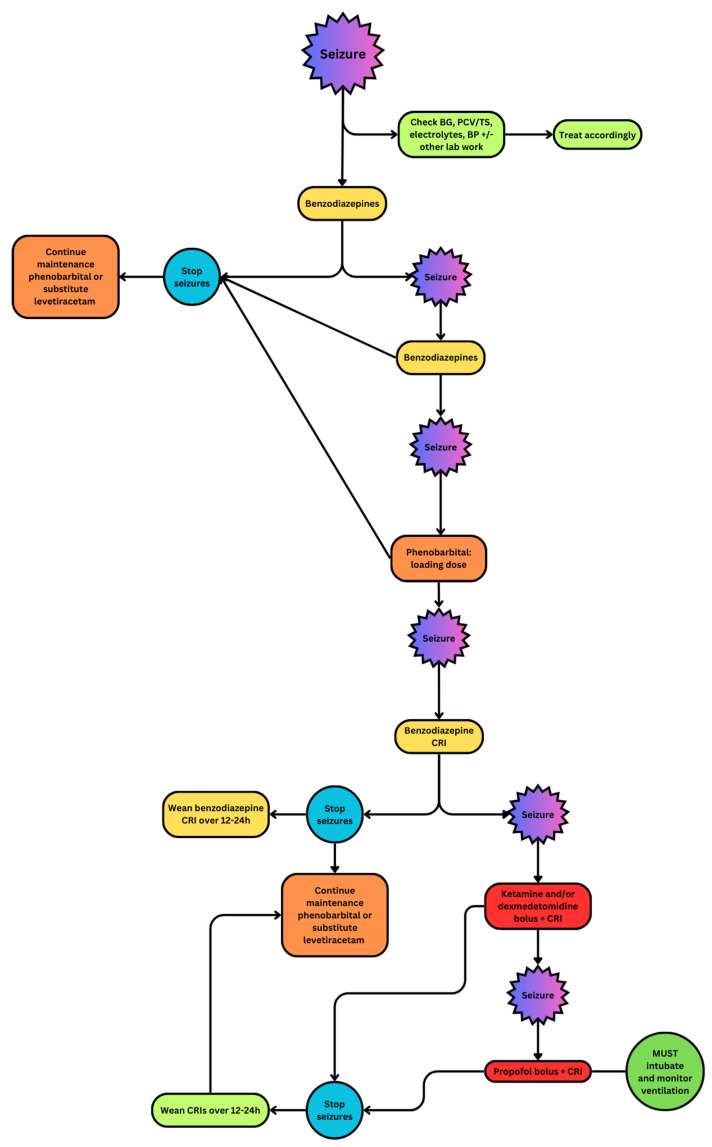
Flowchart demonstrating the use of anticonvulsant drugs in patients with cluster seizures and status epilepticus [12].

## 3. Confirmation and Classification of Seizures

Once the patient has been triaged, the clinician is challenged with distinguishing between true seizures and non-seizure activity that may mimic seizures. The patient presenting to the clinic in status epilepticus is relatively easy to identify as having seizure activity since veterinary professionals have been eyewitnesses to the event. However, most patients with the complaint of “possible seizures” are not actively convulsing at time of exam, leaving owners to interpret “look-alike” syndromes.

When a seizure is suspected, it is important that the clinician is aware of the three most common presentations for seizures (see Table 3). While electroencephalography (EEG) is the gold standard in diagnosing seizure activity, it is not readily available; thus, clinicians need to use the history and physical exam findings and owner event videos to distinguish between true seizures and common look-alikes (see Table 4 and Table 5, and *“Look-alike” Syndromes* Section below). 

## 4. “Look-alike” Syndromes

### 4.1. Syncope

Syncopal events are defined as a sudden and transient loss of consciousness and postural tone originating from short term interruption of perfusion, oxygenation, or delivery of nutrients to the brain [22]. With syncope, some patients may have evidence of myoclonus (a sudden brief, involuntary muscle jerk) [20] or spasms [22] and can urinate or defecate during events; this can make it difficult to differentiate from a generalized seizure. 

Triggering events that lead to impaired oxygenation or nutrient delivery to the brain can occur prior to syncopal episodes. Such events include exercise or exertion, severe coughing, excitement, and even stressors or environmental stimuli [23]. Syncopal patients usually recover almost immediately from the event with no lingering neurologic deficits [22], so assessment for the presence or absence of a post-ictal period is important. Because cardiac disease is a common cause of syncope, further investigation may reveal arrythmias, heart murmurs, or pulse deficits in patients suffering from syncope [22].

### 4.2. Vestibular Disease

Vestibular patients are severely unbalanced, repetitively attempt to “right” themselves if they lose balance to the point of collapse [21], and can appear to be convulsing. However, unlike those having seizures, vestibular patients are aware of their surroundings and responsive to stimuli. They may show clinical signs of vestibular ataxia, postural deficits, strabismus, pathologic nystagmus, and possibly other neurologic deficits related to cranial nerve dysfunction [21].

### 4.3. Tremorgenic Diseases

Tremors can be defined as involuntary, rhythmic oscillatory movements of a body part [20]. Intention tremors seen with cerebellar disease in both species can look similar to the head movements/oscillations noted with seizures; however, unlike with seizure patients, tremoring animals are aware of their surroundings and responsive to stimuli. There are several broad causes of tremors including idiopathic, toxin-induced (e.g., marijuana, bromethalin, metaldehydes, pyrethrins in cats [19]), and metabolic disease (e.g., hypocalcemia). 

### 4.4. Movement Disorders

A movement disorder is an episode of involuntary skeletal muscle contraction during rest or activity in a conscious/mentally appropriate patient [21]. Diagnosis of these episodes is typically based on the look of the episode and signalment of the patient [21]. Examples of movement disorders in dogs include peripheral nerve hyperexcitability disorders, paroxysmal dyskinesia, and dystonia [24].

### 4.5. Behavioral Episodes

Dogs and cats may also suffer from behavioral episodes with various phenotypes which owners may confuse with seizure activity. As with other look-alikes, most patients with behavior disorders are conscious.

## 5. Causes of Seizures

After determining that the event is a seizure, a clinician should have an idea of the causes of seizures. Causes are classified into two large categories based on etiology: intracranial and extracranial (Table 6). As discussed in the *Initial Assessment and Treatment* Section, it is important to diagnose and treat extracranial causes first. Definitive diagnosis of intracranial disease requires advanced diagnostics (e.g., computed tomography, magnetic resonance imaging, cerebrospinal fluid analysis, etc.) [2] that may not be easily available to the emergency clinician or general practitioner. 

There are multiple extracranial causes of seizures that can be ruled out with quick, point-of-care (POC) diagnostics. Whole blood may be collected (ideally during intravenous catheterization) to perform diagnostics such as blood glucose (BG), packed cell volume (PCV), total proteins (TSs), lactate, and/or an electrolyte panel with venous blood gas analysis. A systolic blood pressure reading can rule out severe hypertension. Other diagnostics may include hepatic function testing for hepatic encephalopathy (fasted ammonia levels, pre- and post-prandial bile acid levels), coagulation profiles (PT/aPTT, thromboelastography), and specific infectious disease testing. 

The history, signalment, and physical exam are also helpful in the prioritization of common differentials for seizures. For example, congenital and infectious abnormalities causing seizures are more likely in a young animal, whereas neoplastic or degenerative intracranial diseases are more common in geriatric patients. Another common way for clinicians to quickly remember the most common causes is the mnemonic “DAMNIT-V” [25]. The DAMNIT-V approach to differential diagnoses of seizures is seen in Figure 2 and Table 6. Figure 2 displays the onset and progression of the various differentials over time. For example, an older dog with a primary brain tumor (purple line) may have a late onset of seizure activity that gets worse over time as the tumor grows and as peri-tumoral inflammation causes further structural change. In contrast, a middle-aged dog with an acute onset of seizure activity due to an ischemic event (green line) will initially have more severe signs that will likely improve over time. 

## 6. Other Considerations for Cluster Seizures and Status Epilepticus

Cluster seizures or prolonged status epilepticus can result in detrimental systemic effects. Clinicians should be aware of these possible consequences and monitor at-risk patients for their development. 

### 6.1. Increased Intracranial Pressure and Cerebral Edema

Ongoing and prolonged seizure activity can cause neuronal necrosis which in turn leads to increased inflammatory mediators, permeability of cerebral vessels, and cellular death. This in turn leads to cerebral edema, increasing intracranial pressure and decreasing cerebral blood flow, culminating in cerebral ischemia. Clinicians should be suspicious of increased intracranial pressure when the animal has a decrease in mentation (usually obtundation or stupor) and hypertension (to overcome increased intracranial pressure) with reflexive bradycardia (i.e., the Cushing’s response) [6]. 

Acute treatment of the Cushing’s response is to control the seizures with an AC and treat the intracranial hypertension. Typical treatments include oxygen supplementation without the use of nasal cannulas which can increase intracranial pressure [27], elevation of the head at a 30-degree angle to facilitate outflow of cerebrospinal fluid, and administration of hyperosmolar agents such as mannitol to reduce brain water content [2]. Mannitol (20%) should be given IV through a 0.22 micron in-line filter at a dose of 0.5–1 g/kg over 20 min [6]. Hypertonic saline (7.5% solution 4 mL/kg IV over 5–10 min; 3% solution, 5.4 mL/kg over 5–10 min) [6] can be used in patients that are not severely dehydrated or hypernatremic as it may reduce oxidative stress responses [28]. Another benefit of hypertonic saline is that if given along with isotonic crystalloids, it will provide volume resuscitation. 

### 6.2. Hypoglycemia

Extreme muscle and brain activity may cause hypoglycemia. To treat hypoglycemia, administer a 0.25–1 mL/kg IV bolus of 50% dextrose diluted 1:2–1:4 with an isotonic crystalloid (e.g., 0.9% saline, Lactated Ringer’s Solution). Recheck glucose levels 20–30 min after supplementation to gauge whether further supplementation is required as dextrose is short-acting. If blood glucose levels remain low/low–normal, a constant rate infusion (2.5–5%) should be started. If IV dextrose is not available, apply corn syrup or honey to the mucous membranes; use a tongue depressor in actively seizuring patients to avoid inadvertent bites. Serial monitoring of blood glucose once seizures are controlled may allow the clinician to differentiate whether hypoglycemia is the cause or effect of seizures.

### 6.3. Rhabdomyolysis

Ongoing muscle activity from prolonged seizures may lead to rhabdomyolysis with subsequent myoglobinemia and myoglobinuria. Myoglobinuria can cause renal tubular damage and acute kidney injury. With myoglobinuria, the urine can have a red color to it, and when spun down the color remains the same due to presence of myoglobin. Massive rhabdomyolysis and brain injury will cause elevation of creatinine kinase (CK) [29]. If myoglobinemia and myoglobinuria are present, treat the animal with balanced crystalloid intravenous fluids to promote adequate renal blood flow, glomerular filtration, and diuresis to excrete the myoglobin. Serial chemistry panels including CK levels and urinalysis at least once a day will show when rhabdomyolysis is resolved and identify/chart the progression of any kidney disease. Perform serial laboratory testing of serum renal values, electrolytes, and urinalyses at least every 24 h. Creatine kinase activity increases rapidly with peak concentrations at 2–12 h post-injury and may normalize to within reference range 24–48 h after injury has halted [29,30]. Ongoing CK elevations should prompt investigation for underlying myopathies or subtle uncontrolled/ongoing seizure activity.

### 6.4. Hyperlactatemia

During the seizure event, brain, heart, and skeletal muscle oxygen demand rises but patients are not always able to fully compensate for this increase in oxygen demand. This causes a type A hyperlactatemia; if seizures are controlled, this hyperlactatemia typically resolves on its own within an hour without specific interventions [31]. Persistent hyperlactatemia once seizures are controlled might indicate other underlying causes of shock/disease. 

### 6.5. Hyperthermia

Prolonged muscle contraction can result in hyperthermia. Severe hyperthermia (rectal temperature >106 °F or 41 °C) can lead to heat stroke and damage to many vital organ systems including the gastrointestinal tract, liver, kidneys, brain, heart, and coagulation system. Active cooling (fans, water bath, IV fluids) until the rectal (core) temperature is ~103 °F (39.4 °C) is warranted. Cooling beneath this core temperature may lead to rebound hypothermia. This occurs when cold peripheral blood replaces centrally located blood, dropping the core temperature. If the rectal temperature is still elevated after control of seizures and cooling, consider other causes of a fever which may or may not be related to seizures. Should the patient’s seizures require total intravenous anesthesia (e.g., propofol constant rate infusion), breathing high-flow or non-rewarmed air can also be a method of cooling. In some cases, persistent or recurring temperature spikes can be a sign of subtle seizure activity. 

### 6.6. Neurogenic Pulmonary Edema

While uncommon, a type of non-cardiogenic pulmonary edema termed neurogenic pulmonary edema (NPE) can arise after severe or ongoing seizure activity. The increase in intracranial pressure during a seizure leads to a surge in catecholamines, resulting in vasoconstriction and increased vascular permeability [32]. The resulting alveolar capillary fluid leakage causes pulmonary edema [6,32]. Diagnosis of neurogenic pulmonary edema is made from thoracic radiographs. A bilaterally symmetric interstitial-to-alveolar pattern most prominent in the caudodorsal lung fields is commonly seen but may take upwards of 24 h to form (see Figure 3). Thoracic radiographs may also be useful to serve as a baseline for patients at high risk of developing non-cardiogenic pulmonary edema secondary to seizures, and they may also be used to rule out evidence of aspiration pneumonia/pneumonitis and evidence of primary or malignant neoplasia of the thoracic cavity. Unlike with cardiogenic pulmonary edema (CPE), the use of diuretics such as furosemide is controversial [33] and may not be recommended as a first-line treatment with NPE as volume overload is not present in NPE as it is with CPE. While there have been speculations that furosemide causes vasodilation leading to improved ventilation/perfusion mismatch, studies showing its consistent use for NPE are lacking [32]. There is also skepticism of furosemide’s benefit as it likely leaves behind leaked proteins and does not necessarily prevent further vascular leakage of fluid. Treatment relies on supplemental oxygen as needed, and in severe cases, mechanical ventilation may be required [32].

## 7. Clinical Approach to Seizures

When patients are presented to the clinic for seizure activity, they usually present in one of three ways (see Table 3):1.A single seizure at home, no active seizure at presentation.2.A cluster of seizures, no active seizure at presentation.3.In status epilepticus or actively seizuring.

For all three presentations, once the clinician determines that the episode is consistent with seizure activity, the goals of the visit include stopping any ongoing seizures, immediate identification and treatment of extracranial causes of seizures (e.g., hypoglycemia), and, in some cases, initiation of long-term AC therapy. There are no strict recommendations on when to start maintenance AC drugs in patients with single seizures; the authors typically recommend these pets be started on maintenance AC drugs if there are more than two seizures in one month, if they have cluster seizures, or in those that present in status epilepticus. Patients that present with cluster seizures or in status epilepticus should be started and continued on AC therapy and ideally be hospitalized for at least 24 h to ensure seizures have stopped [12]. For specific information about monitoring obtunded, stuporous, or comatose patients, the authors recommend referring to the “Neurological status” chapter in *Monitoring and intervention for the critically ill small animal: the rule of 20* [34]. The authors refer the reader to other sources regarding long-term/maintenance AC drug doses and options [2,35,36,37]. Cluster seizure or SE patients should be started on emergency and maintenance AC drugs even if they have not been evaluated by a veterinary neurologist and/or if advanced diagnostics to pursue an underlying diagnosis have been declined.

## 8. Conclusions

Seizures are a very common disease process for which dogs and cats present to primary or emergency centers. Immediate identification, triage, and treatment of status epilepticus or seizure activity is important for both emergency and general practitioners. Use of point of care diagnostics is imperative to identify metabolic causes and sequelae of seizures which should be treated as an emergency. Clinicians should be comfortable differentiating “look-alikes” from seizures, know when to administer first- and second-line ACs, and be able to identify and treat common consequences of prolonged seizure activity. 

## Figures and Tables

**Figure 2 vetsci-11-00277-f002:**
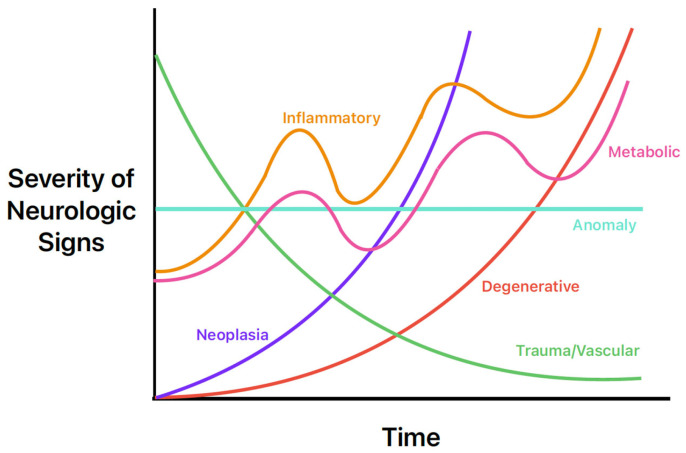
Neurologic sign severity according to major differential diagnoses of seizures in dogs and cats. Image adapted from Neurological Examinations: Localization and Grading [26].

**Figure 3 vetsci-11-00277-f003:**
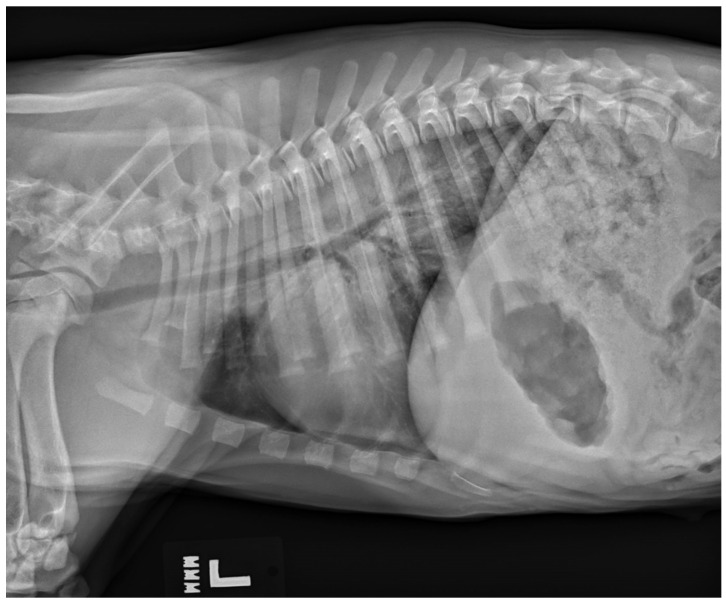
A left lateral thoracic radiograph showing the classic caudodorsal interstitial-to-alveolar pulmonary pattern of a patient with neurogenic pulmonary edema.

**Table 1 vetsci-11-00277-t001:** Treatment of common metabolic causes of seizures in dogs and cats.

Cause	Clinical Identification	Treatment	Comments
Hypoglycemia	BG <3.3 mmol/L [1]	0.5–1 mL/kg 50% dextrose diluted 1:2–1:4 with an isotonic crystalloid [1,2]	Recheck BG 20–30 min after bolus Repeat boluses as necessary Can start a CRI once bolus therapy meets normal range
Liver failure	Hyperammonemia Abnormal biochemical markers of liver function (↓ BG, ↓ BUN, ↓ albumin, ↑ TBIL, ↓ cholesterol) Elevated bile acids	1–10 mL/kg (3 parts lactulose, 7 parts warm water) as retention enema for 20–30 min [3]	Can repeat q6-8hr until patient able to take oral lactulose
Erythrocytosis	PCV >65%	IV isotonic fluid boluses and/or aggressive rate of fluid therapy until PCV <60% Consider phlebotomy if erythrocytosis not due to decreased intravascular fluid volume	
Severe anemia	PCV <10%	Fresh whole blood or packed red blood cell transfusion	
Severe uremia	Azotemia on chemistry profile or reagent strip (e.g., Azostix®)	IV fluid diuresis or hemodialysis if indicated	Monitor kidney values (BUN, creatinine), electrolytes, and body weight at least q24h
Hypocalcemia	TCa <1.5 mmol/L iCa <0.8 mmol/L [4]	0.5–1.5 mL/kg 10% calcium gluconate given over 20–30 min	Monitor ECG during administration Rapid administration can cause hypotension, bradycardia, arrhythmias
Hyperthermia	Rectal temperature >106 °F (>41 °C)	Active cooling: water bath (tap water), fans, IV fluid therapy	Stop cooling efforts once rectal temperature reaches 103 °F (39.4 °C)
Hypertension	SBP >180 mmHg	Acute drugs to treat hypertension (e.g., amlodipine, nitroprusside, etc.) [5]	Serial measurements are recommended. Rule out hypertension due to pain, stress/excitement, or Cushing’s response

BG: blood glucose; CRI: constant rate infusion; TBIL: total bilirubin; PCV: packed cell volume; BUN: blood urea nitrogen; TCa: total calcium; iCa: ionized calcium; SBP: systolic blood pressure; IV: intravenous; ECG: electrocardiogram.

**Table 2 vetsci-11-00277-t002:** Anticonvulsant drugs for the emergent treatment of cluster seizures/status epilepticus in dogs and cats [6].

Drug	Dose	Comments
First-line ACs		
Diazepam	IV: 0.5 mg/kg, repeat 2–3 times as necessary CRI: 0.5 mg/kg/hr for 12–24 h or at least until maintenance drug is appropriately loaded PR: 1 mg/kg	Owners can be trained to administer PR at home. PR may not reach therapeutic levels [7,8] Do not administer PO in cats; risk of hepatic failure [9,10,11] CRI can be started if seizures recur after 2 doses of benzodiazepines given but after phenobarbital is started
Midazolam	IV: 0.3 mg/kg, repeat 2–3 times as necessary CRI: 0.3–0.4 mg/kg/hr for 12–24 h or at least until maintenance drug is appropriately loaded IN: 0.3 mg/kg (with or without atomizer) IM: 0.3 mg/kg	Preferred over diazepam [12] Owners can be trained to administer IN at home PR route not recommended in dogs due to poor bioavailability and subtherapeutic effects [8]. Information in cats is lacking. CRI can be started if seizures recur after 2 doses of benzodiazepines given but after phenobarbital is started
Second-line ACs		
Phenobarbital	4 mg/kg IV every 20–30 min for a total of 16–20 mg/kg (SE loading dose) 4 mg/kg IV every 2–4 h for a total of 16–20 mg/kg (clusters loading dose)	Can induce anesthesia; must monitor patient vitals, oxygenation, and ventilation status Contraindicated in patients with liver disease or respiratory disease (high risk of sedation and respiratory depression) [13] PO should only be given if patient able to swallow
Levetiracetam	60 mg/kg IV/PO loading dose (can redose 3–4 times in a 24 h period to achieve seizure control)	Shown to be useful as part of multi-modal AC protocol for SE alongside benzodiazepines and phenobarbital [14,15] PO should only be given if patient able to swallow
Third-line ACs		
Ketamine	1–5 mg/kg IV bolus, then 1 mg/kg/hr CRI [16]	Can be used alone or concurrently with dexmedetomidine CRI [12] Use if refractory to benzodiazepines, levetiracetam, and phenobarbital
Dexmedetomidine	3 ug/kg IV bolus, then 3–7 ug/kg/hr CRI [16]	Can be used alone concurrently with ketamine CRI [12] Use if refractory to benzodiazepines, levetiracetam, and phenobarbital
Propofol	2–8 mg/kg IV slow (25% of total dose every 30 s until desired effect achieved)	Maintain anesthesia with CRI: 0.1–0.4 mg/kg/min [17] Must monitor patient’s vitals, oxygenation, and ventilation status very closely Intubate all animals receiving propofol

AC: anticonvulsants; CRI: constant rate infusion; IM: intramuscular; IN: intranasal; IV: intravenous; PR: per rectum; SE: status epilepticus.

**Table 3 vetsci-11-00277-t003:** Definitions of the most common seizure types in dogs and cats.

Event	Definition
Seizure	Sudden, short-lasting, and transient events characterized by motor, autonomic, or behavioral features, or some combination of these [18]
Cluster seizures	Two or more seizures within a 24 h period [19]
Status epilepticus	Continuous seizure activity lasting longer than 5 min, or greater than one sequential seizure without full recovery of consciousness in between seizures, with a duration of greater than 30 min [12,18]

**Table 4 vetsci-11-00277-t004:** Questions to ask pet owners who present their dog or cat for a suspected seizure.

Important Questions to Ask Pet Owners with Suspected or Confirmed Seizure Activity
How would you describe the event? How long did the event last (in minutes)? How many episodes occurred? Did your pet urinate or defecate during the episode? Was your pet responsive/aware or did it lose consciousness during the episode? Do you think anything triggered this event? How long did it take for your pet to return to what you perceive is normal? Is this the first time your pet has had this type of episode? ○ If not, how many times has it happened before? ○ When was the most recent episode preceding today’s episode? What was your pet doing prior to this event? After? Is your pet on any medications? ○ If so, which medications and when were they last given? Does your pet have any underlying diseases or diagnoses? Does your pet have a history of seizures? ○ When were they diagnosed and what was the proposed cause? ○ When was the last known seizure? ○ What is the frequency of seizures? ○ How does this event compare to previous seizures? ○ Is your pet on any anti-seizure medications? If so, which ones and what dose/frequency? Is there any way your pet could have gotten into any toxins (marijuana, xylitol-containing foods, rat/mice bait, snail bait, antifreeze, etc.)? Do you have a video of the episode?

**Table 5 vetsci-11-00277-t005:** Characteristics of seizures and the most common “look-alike” conditions.

Disease Process	During Event	Peri-Event	During Event
Seizures	Dogs: Focal vs. generalized rhythmic movements	May have pre-ictal behavior(s) Generalized seizures should typically have a post-ictal phase of abnormal mentation/behavior that may last minutes to days Focal seizures sometimes do not have post-ictal behavior	Loss of consciousness May have autonomic signs (urination, defecation, salivation)
	Cats: tonic-clonic movements often accompanied by explosive/unpredictable muscular movements (jumping), growling, chewing at the tail, hypersalivation, mydriasis, facial twitching, chomping		
Syncope	Unconscious or weak for a short period, often a loss of muscle tone, diffusely	Normal mentation before, sometimes during, and shortly (seconds to minutes) after event Typically, do not have existing post-event behavior or clinical signs	Can be associated with increased activity or stressor prior to event May urinate May not completely lose consciousness during event
Vestibular disease	Unsteady, rapid, and abrupt loss of balance ± falling to one or both sides, ± nystagmus and head tilt, ± hypersalivation	May have ongoing or inter-episode vestibular signs (head tilt, nystagmus)	Aware, responsive to stimuli When attempting to “right” themselves, may paddle limbs as if seizuring May hypersalivate if feeling dizzy
Tremors	Rhythmic oscillations without loss of consciousness [20]	Normal prior to onset	Aware, responsive to stimuli
Movement Disorders	Sudden and involuntary movements or spasms [21]	Neuro exam normal between episodes	No loss of consciousness during episodes

**Table 6 vetsci-11-00277-t006:** Common intracranial and extracranial causes of seizures in dogs and cats. This list is not exhaustive but displays the DAMNIT-V scheme of organizing differentials into Degenerative, Anomalous, Metabolic, Neoplastic, Inflammatory/Infectious, Trauma, and Vascular.

Intracranial	Extracranial
Degenerative: Thiamine deficiency (in cats) Specific enzyme deficiencies Anomalous: Hydrocephalus Metabolic: Storage diseases Neoplastic: Meningioma Glioma Lymphoma Metastatic disease Inflammatory/Infectious: Immune mediated (meningoencephalitis of unknown etiology) Infectious (viral, protozoal, bacterial, fungal, rickettsial) Trauma/toxin: Traumatic brain injury Bromethalin Vascular: Ischemia Hemorrhage	Degenerative: Severe hyperthermia Metabolic: Hypoglycemia Uremia Hyperlipidemia Hepatic encephalopathy Electrolyte abnormalities (Na, Ca) Hyperosmolar disorders (e.g., hyperosmolar hyperglycemic syndrome) Urea cycle toxicity (rare) Toxicities: Lead, organophosphates, chlorinated hydrocarbons, xylitol, caffeine, ethylene glycol Vascular: Blood dyscrasias (anemia, severe thrombocytopenia) Hypertension Hyperviscosity (multiple myeloma, polycythemia, erythrocytosis)

Na: sodium; Ca: calcium.

## Data Availability

No new data were created or analyzed in this article. Data sharing is not applicable to this article.

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
