# Peer review of "Emergency Approach to Acute Seizures in Dogs and Cats"

_vetsci, 2024, doi:10.3390/vetsci11060277_

Round 1
Reviewer 1 Report
Comments and Suggestions for Authors
Congratulations for the paper.
Overall, it is very well written and structured in my opinion.
I have a few comments and considerations to make which I hope will enhance the final quality of the paper.

Author Response
|
Reviewer Comment |
Response |
|
Table 2. The absorption and consequent sedative effect by PR of benzodiazepines (particularly midazolam) in dogs has been mentioned in appropriate literature sources. In the cat, no. However, there is a reference to the use of midazolam by PR. This reference is in line with reports of poor absorption of midazolam by the PR route. I would recommend adding it as a reference. |
Thank you for this feedback. The only article we could find in the primary literature regarding cats and PR midazolam/benzodiazepines is regarding sedation effects and parameters, thus we have added the line “Information in cats is lacking” in Table 2.
|
|
Line 159-161: why the text is underlined? |
Thank you for catching this. This was not intended to make the final draft for review and thus has been corrected (underline removed). |
|
Line 212-216: In the crisis management scheme, you do not mention ventilation and ventilation associated with propofol TIVA (which you do mention). It would be interesting to discuss, whether you would recommend ventilation and if so, whether hyperthermia can also be managed with ventilation (fresh gas flows) and drug-induced vasodilation, such as halogenates. |
Thank you for this insight and consideration. While your suggestion is valid, in the current emergency and critical care literature, ventilation is not a practice used to cool patients; a, b however, we will add a brief note of this which can now be found in lines 222-224.
|
a: Bruchim Y, Kelmer E. Canine heat stroke. In: Drobatz KJ, Hopper K, Rozanski E, and Silverstein DC, eds. Textbook of Small Animal Emergency Medicine. Volume 2. John Wiley & Sons; 2019: 942-949.
b: Mazzaferro EM. Heat stroke and heat-induced illness. In: Blackwell’s Five-Minute Veterinary Consult Clinical Companion: Small Animal Emergency and Critical Care. Blackwell Publishing Ltd; 2010: 319-328.
Reviewer 2 Report
Comments and Suggestions for Authors
The paper is well written, of great scientific depth and of sincere and valid help. Thanks to all the authors for the research
Author Response
|
Reviewer Comment |
Response |
|
The paper is well written, of great scientific depth and of sincere and valid help. Thanks to all the authors for the research |
Thank you for your time and review of this article. We hope that this will serve as a practical and succinct guide to the emergent management of seizures in cats and dogs. |
Reviewer 3 Report
Comments and Suggestions for Authors
The authors aimed to present a review about the emergency management of seizures in small animals. The subjetc is very relevant and can provide useful and updated information for the clinical veterinarian. The manuscript is very written. The references are updated. I have some minor suggestions to improve the presentation of the manuscript in the journal:
a) Page 2; title of Table 1: ...seizures in dogs or cats ? Or both ?
Inside of this table. Severe uremia - Comments: "Monitor kidney values" - BUN, creatinine or both ? Specify
b) Page 8, line 102. "Cerebellar disease can also look similar to seizures" - Briefly, please specify the differences.
c) Pag 11, lines 196-197: "Creatine kinase activity increases..." - Is this the cause of seizure ? Please, explain the pathogenesis
d) Pag. 12, lines 231-232: "Unlike cardiogenic pulmonary edema...with NPE". - Why ? Please explain.
Author Response
|
Reviewer Comment |
Response |
|
a) Page 2; title of Table 1: ...seizures in dogs or cats ? Or both ? Inside of this table. Severe uremia - Comments: "Monitor kidney values" - BUN, creatinine or both ? Specify
|
Thank you for your remarks.
Line 53: We have added “in Dogs and Cats” to the title of Table 1
Table 1 “Severe uremia”: We have added in parenthesis “BUN, creatinine”. |
|
b) Page 8, line 102. "Cerebellar disease can also look similar to seizures" - Briefly, please specify the differences. |
Thank you for pointing out that the sentence was confusing. We have re-written this sentence as follows to hopefully eliminate any confusion: “Intention tremors seen with cerebellar disease in both species can look similar to head movements/oscillations noted with seizures...” starting in line 104-105. |
|
c) Pag 11, lines 196-197: "Creatine kinase activity increases..." - Is this the cause of seizure ? Please, explain the pathogenesis |
Creatinine kinase is released with muscle damage, as seen with rhabdomyolysis. Lines 201-203 describes clinically how long CK elevations are expected after muscle damage has occurred. CK is not a cause of seizures and is a biomarker of muscle damage. Added “from prolonged seizures” to line 191 to emphasize this is a result, not a cause of seizure activity. |
|
d) Pag. 12, lines 231-232: "Unlike cardiogenic pulmonary edema...with NPE". - Why ? Please explain. |
Thank you for this question. We have added in a brief explanation on our opinion on the use of diuretics in NPE, now seen in lines 238-244. |
Reviewer 4 Report
Comments and Suggestions for Authors
This is a very basic review of seizures and potential rule outs and emergent treatments available to first responders.
I am concerned that this paper looks very similar to Journal of Veterinary Emergency and Critical Care 27(3) 2017, pp 288–300 doi: 10.1111/vec.12604, albeit, much more simplistic in its presentation.
My expectation was to have the manuscript focus on the emergency management of seizures, but I found this manuscript focuses more on the initial diagnostics and rule outs. This is not a bad thing, maybe reconsider the title and the purpose of the paper.
The manuscript tables are helpful and easy to find information, but figure 1 is very similar to Journal of Veterinary Emergency and Critical Care 27(3) 2017, pp 288–300 doi: 10.1111/vec.12604. I did like the common historical questions table.
This paper lacks the deep dive into the anticonvulsant medications utilized in the ER. There are many reasons for drug choice which is overlooked (i.e. diazepam vs midazolam). No discussion of MOA or side effects. Other secondary drugs are excluded (i.e. zonisamide). Discussion on patient stabilization and nursing care needs to be expanded. Prognosis is not discussed.
Author Response
|
Reviewer Comment |
Response |
|
I am concerned that this paper looks very similar to Journal of Veterinary Emergency and Critical Care 27(3) 2017, pp 288–300 doi: 10.1111/vec.12604, albeit, much more simplistic in its presentation. |
Thank you for your review. While the authors agree there is some overlap between this manuscript and the mentioned JVECC article, the goal of the current manuscript is to provide a broad overview of the emergent approach to single, cluster, and SE seizure presentations in cats and dogs rather than just SE. The authors were aiming for very accessible information to the general practice and ER veterinarian audience, which is why much of the information is in table/graphical format. |
|
My expectation was to have the manuscript focus on the emergency management of seizures, but I found this manuscript focuses more on the initial diagnostics and rule outs. This is not a bad thing, maybe reconsider the title and the purpose of the paper. |
Thank you for your comment; we have changed the title and the abstract to better reflect our manuscript. |
|
The manuscript tables are helpful and easy to find information, but figure 1 is very similar to Journal of Veterinary Emergency and Critical Care 27(3) 2017, pp 288–300 doi: 10.1111/vec.12604. I did like the common historical questions table. |
Thank you for pointing out that we did not cite the origin of Figure 1. Figure 1 was created based on the 2024 ACVIM Consensus Statement.a Any similarity to Figure 1 in the JVECC article was unintentional, but it is likely that the consensus statement drew from the JVECC article. Thank you for your positive feedback on Table 4. |
|
This paper lacks the deep dive into the anticonvulsant medications utilized in the ER. There are many reasons for drug choice which is overlooked (i.e. diazepam vs midazolam). No discussion of MOA or side effects. Other secondary drugs are excluded (i.e. zonisamide). |
We understand and respect the reviewer’s comments, however, the goal of this paper is to provide a stepwise “how to” approach with less weight on mechanisms and more weight on emergent treatment. We have added a comment in Table 2 indicating the consensus statement’s recommendation of using midazolam over diazepam.a
The authors have chosen not to discuss the use of anticonvulsants considered by the ACVIM consensus statement for maintenance only (e.g. zonisamide).a
The authors urge the readers to delve deeper into specifics found in other resources, now including your referenced/recommended article from JVECC, for long term management which are listed in lines 268-269. |
|
Discussion on patient stabilization and nursing care needs to be expanded. Prognosis is not discussed. |
Thank you for bringing up nursing care. The authors considered including a section about this but decided not to for two reasons. First, in the authors’ experience, the vast majority of hospitalized seizure patients are normal or almost normal in hospital. Secondly, given that the goal of the paper is how to assess, categorize, and initially treat a patient with seizures, we felt that an extensive discussion of nursing care was beyond the scope and length requirements of this article. The paper includes multiple brief mentions of nursing care, specifically lines 49-52, 164-167, 182-184, and 217-218. We have also added an additional reference for interested readers in lines 265-268.
Likely due to the euthanasia bias mentioned in the 2017 JVECC paper, the ACVIM consensus statement avoids drawing conclusions about prognosis aside from mentioning that the use of an injectable or inhalant anesthetic is associated with a poor long-term outcome.a Therefore, the authors have chosen not to discuss prognosis in this article. |
a: M. Charalambous, K. Muñana, E. E. Patterson, S. R. Platt, and H. A. Volk, “ACVIM Consensus Statement on the management of status epilepticus and cluster seizures in dogs and cats,” J Vet Intern Med, vol. 38, no. 1, pp. 19–40, 2024, doi: 10.1111/jvim.16928.